# MolEBM: Molecule Generation and Design by Latent Space Energy-Based Modeling

## Abstract

Generation of molecules with desired chemical and biological properties such as high drug-likeness, high binding affinity to target proteins, is critical in drug discovery. In this paper, we propose a probabilistic generative model to capture the joint distribution of molecules and their properties. Our model assumes an energy-based model (EBM) in the latent space. Given the latent vector sampled from the latent space EBM, both molecules and molecular properties are conditionally sampled via a molecule generator model and a property regression model respectively. The EBM in a low dimensional latent space allows our model to capture complex chemical rules implicitly but efficiently and effectively. Due to the joint modeling with chemical properties, molecule design can be conveniently and naturally achieved by conditional sampling from our learned model given desired properties, in both single-objective and multi-objective optimization settings. The latent space EBM, molecule generator, and property regression model are learned jointly by approximate maximum likelihood, while optimization of properties is accomplished by gradual shifting of the model distribution towards the region supported by molecules with high property values. Our experiments show that our model outperforms state-of-the-art models on various molecule design tasks.

## 1 Introduction

In drug discovery, it is of vital importance to find or design molecules with desired pharmacologic or chemical properties such as high drug-likeness and binding affinity to a target protein. It is challenging to directly optimize or search over the drug-like molecule space since it is discrete and enormous, with an estimated size is on the order of $10^{33}$ (Polishchuk et al., 2013).

Recently, a large body of work attempts to tackle this problem. The first line of work leverages deep generative models to map the discrete molecule space to a continuous latent space, and optimizes molecular properties in the latent space with methods like Bayesian optimization (Gómez-Bombarelli et al., 2018; Kusner et al., 2017; Jin et al., 2018). The second line of work recruits reinforcement learning algorithms to optimize properties in the molecular graph space directly (You et al., 2018; De Cao & Kipf, 2018; Zhou et al., 2019; Shi et al., 2020; Luo et al., 2021). A number of other efforts have been made to optimize molecular properties with genetic algorithms (Nigam et al., 2020), particle-swarm algorithms (Winter et al., 2019) specialized MCMC methods (Xie et al., 2021).

In this work, we propose a method along the first line mentioned above, by learning a probabilistic latent generative model of molecule distributions and optimizing chemical properties in the latent space. Given the central role of latent variables in this approach, we emphasize that it is critical to learn a latent space model that captures the data regularities of the molecules. Thus, instead of assuming a simple Gaussian distribution in the latent space as in prior work (Gómez-Bombarelli et al., 2018; Jin et al., 2018), we assume a flexible and expressive energy-based model (EBM) (LeCun et al., 2006; Ngiam et al., 2011; Kim & Bengio, 2016; Xie et al., 2016; Kumar et al., 2019; Nijkamp et al., 2019; Du & Mordatch, 2019; Grathwohl et al., 2019; Finn et al., 2016) in latent space. This leads to a *latent space energy-based model* (LSEBM) as studied in Pang et al. (2020); Nie et al. (2021), where LSEBM has been shown to model the distributions of natural images and text well. For molecule modeling, without any explicit validity constraints in generation, our model generates molecules with high validity with simple SMILES representation (Weininger, 1988).

Given our goal of property optimization, we learn a joint distribution of molecules and their properties. Our model consists of 1) an EBM in a low-dimensional continuous latent space, 2) a generator mapping from the latent space to the observed molecule space, and 3) a property regression model mapping from the latent space to the property values (see Figure 1). We call our model as *MolEBM*. All three components in our model are learned jointly by approximate maximum likelihood. A learned model generates a molecule with a high property value in two steps: 1) given the property value, sample the latent vector; 2) given the sampled latent vector, generate a molecule (see the top-to-bottom path in Figure 1a). Since the learned model approximates the data distribution, directly sampling from the learned model conditional on a high property value does not work well since a molecule with a high property value is most likely not in the original data distribution. We thus design a method to gradually shift the learned distribution towards the region supported by molecules with high property values, and sample molecules with desirable properties from the shifted distribution.

In drug discovery, most often we need to consider multiple properties simultaneously. Our model can be extended to this setting straightforwardly. With our method, we only need to add a regression model for each property, while the learning and sampling methods remain the same. Learning the model involves inferring the latent vector given both the molecule and the property value, and we recruits Langevin dynamics instead of amortized inference network for inference computation. This design makes our approach versatile in dealing with varying number of properties.

We evaluate our method in various settings including single-objective optimization and multi-objective optimization. Our method outperforms prior methods by significant margins.

In summary, our contributions are as follows:

- We propose to learn a latent space energy-based model for the joint distribution of molecules and molecular properties.
- We develop a *sampling with gradual distribution shifting* method, enabling us to extrapolate the data distribution and sample from the region supported by molecules with high property values.
- Our methods are versatile enough to be extended to optimizing multiple properties together.
- Our model achieves state-of-the-art performances on a wide range of molecule optimization tasks.

## 2 RELATED WORK

**Optimization with Generative Models.** Deep generative models approximate the distribution of molecules with desired biological or non-biological properties. Existing approaches for generating molecules include applying variational autoencoder (VAE) (Kingma & Welling, 2014) and generative adversarial network (GAN) (Goodfellow et al., 2014) etc. to molecule data (Gómez-Bombarelli et al., 2018; Jin et al., 2018; De Cao & Kipf, 2018; Honda et al., 2019; Madhawa et al., 2019; Shi et al., 2020; Zang & Wang, 2020; Kotsias et al., 2020; Chen et al., 2021; Fu et al., 2020; Liu et al., 2021; Bagal et al., 2021; Eckmann et al., 2022; Segler et al., 2018). After learning continuous representations for molecules, they are further able to optimize using different methods. (Segler et al., 2018) proposes to optimize by simulating design-synthesis-test cycles. (Gómez-Bombarelli et al., 2018; Jin et al., 2018; Eckmann et al., 2022) propose to learn a surrogate function to predict properties, and then use Bayesian optimization to optimize the latent vectors. However, the performance of this latent optimization is not satisfactory due to three major issues. First, it is difficult to train an accurate surrogate predictor especially for those novel molecules with high properties along the design trajectories. Second, as the learned latent space tries to cover the fixed data space, its ability to explore the targets out of the distribution is limited (Brown et al., 2019; Huang et al., 2021). Third, those methods are heavily dependent on the quality of learned latent space, which requires non-trivial efforts to design encoders when dealing with multiple properties. To address above issues, (Eckmann et al., 2022) use VAE to learn the latent space and train predictors separately using generated molecules, and then leverage latent inceptionism, which involves the decoder solely, to optimize the latent vector with multiple predictors. In this paper, we propose an encoder-free model in both training and optimization to learn the joint distribution of molecules and properties, and make it possible to obtain several adequate predictors. We then design an efficient algorithm to shift the learned distribution iteratively.

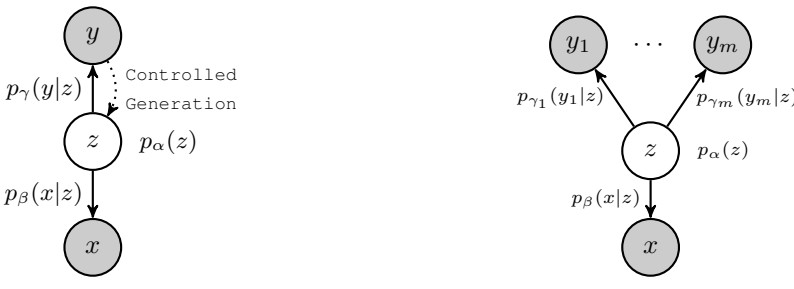

(a) Single-Objective Optimization.  (b) Multi-Objective Optimization.

Figure 1: An illustration of MolEBM. $x$ represents a molecule, $z$ is the latent vector, $y$ is a molecular property of interest, $\{y_j\}_{j=1}^m$ indicates $m$ properties.

**Optimization with Reinforcement Learning and Evolutionary Algorithms.** Reinforcement learning (RL) based methods directly optimize and generate molecules in an explicit data space (You et al., 2018; Zhou et al., 2019; Jin et al., 2020; Gottipati et al., 2020). By formulating the property design as a discrete optimization task, they can modify the molecular substructures guided by an oracle reward function. However, the training of those RL-based methods can be viewed as rejection sampling which is difficult and inefficient due to the random-walk search behavior in the discrete space. Evolutionary algorithms (EA) also formulate the optimization in a discrete manner (Nigam et al., 2020; Jensen, 2019; Xie et al., 2021; Fu et al., 2021a;b). By leveraging carefully-crafted combinatorial algorithms, they can search the molecule graph space in a flexible and efficient way. However, the design of those algorithms is non-trivial and domain specific.

## 3    METHODS

### 3.1    PROBLEM SETUP AND OVERVIEW

We use the SELFIES representation for molecules (Krenn et al., 2020). It encodes each molecule as a string of characters and ensures validity of all SELFIES strings. Let $x = (x^{(1)}, ..., x^{(t)}, ..., x^{(T)})$ be a molecule string encoded in SELFIES, where $x^{(t)} \in \mathcal{V}$ is the $t$-th character and $\mathcal{V}$ is the vocabulary. Suppose $y \in \mathbb{R}$ represents a molecular property of interest. Then the problem we attempt to tackle is to optimize $x$ such that its property $y = y^*$ where $y^*$ is some desirable value for $y$. We take a probabilistic approach and treat the optimization problem as a sampling problem, that is,

$$x^* \sim p(x|y = y^*). \tag{1}$$

This is a *single-objective optimization* problem since only one property is targeted. In real-world drug design settings, we are more likely to need to optimize multiple properties simultaneously, that is, *multi-objective optimization*. Suppose we optimize for $\{y_j \in \mathbb{R}\}_{j=1}^m$, then our task is to sample,

$$x^* \sim p(x|y_1 = y_1^*, ..., y_m = y_m^*). \tag{2}$$

To address these problems, we propose a solution under a unified probabilistic framework. As a first step, we need to model the data distribution of molecules, $p_{\text{data}}(x)$. To this end, we recruit latent space energy-based model (Pang et al., 2020; Nie et al., 2021) for its expressiveness. LSEBM assumes latent vector $z \in \mathbb{R}^d$ in a low dimensional latent space follows an energy-based model, while the observed $x$ is generated by a generator conditional $z$, that is, $p(x, z) = p(x|z)p(z)$. We introduce LSEBM within the context of molecule data in §3.2.

For the purpose of property optimization, we propose to model the joint distribution of molecules and molecular properties (§3.3). We first consider the single-objective optimization problem (eq. 1). Given the expressiveness of LSEBM, we assume that the latent vector $z$ captures data regularities in $x$ (evaluated in §4.2 and §A.4). Thus, we can learn a simple regression model from the low-dimensional latent space, that is, $p(y|z)$, and the joint distribution is $p(x, y, z) = p(z)p(x|z)p(y|z)$. The model is learned with maximum likelihood (see §3.4 and Algorithm 1).

With the learned model, we can optimize $x$ given $y = y^*$ by ancestral sampling $z^* \sim p(z|y = y^*)$ and $x^* \sim p(x|z = z^*)$. However, if $y^*$ deviates from the observed data distribution of $y$, this naive

solution involves sampling in an extrapolated regime (or out of distribution regime) where $y^*$ is not in the effective support of the learned distribution. We propose a *Sampling with Gradual Distribution Shifting* (SGDS) approach where we 1) sample on the boundary of the effective support of the learned distribution, and 2) gradually shift the learned distribution with these boundary samples to a region where it is supported by high property values (see §3.5 and Algorithm 2).

Our model is designed to be versatile such that it admits straightforward extension to multi-objective optimization. To optimize $x$ given $\{y_j = y_j^*\}_{j=1}^m$, we can simply augment the joint distribution with more regression models, i.e., $p(x, z, y_1, ..., y_m) = p(z)p(x|z)\prod_{j=1}^m p(y_j|z)$. The sampling procedure follows the same SGDS approach. See §3.6 for more details on multi-objective optimization.

## 3.2 LATENT SPACE ENERGY-BASED MODEL

Suppose $x = (x^{(1)}, ..., x^{(t)}, ..., x^{(T)})$ is a molecule string in SELFIES and $z \in \mathbb{R}^d$ is the latent vector. Consider the following model,

$$z \sim p_\alpha(z), \quad x \sim p_\beta(x|z), \tag{3}$$

where $p_\alpha(z)$ is a prior model with parameters $\alpha$, and $p_\beta(x|z)$ is a generation model with parameters $\beta$. In VAE (Kingma & Welling, 2014), the prior is simply assumed to be an isotropic Gaussian distribution. In our model, $p_\alpha(z)$ is formulated as an energy-based model,

$$p_\alpha(z) = \frac{1}{Z(\alpha)} \exp(f_\alpha(z))p_0(z), \tag{4}$$

where $p_0(z)$ is a reference distribution, assumed to be isotropic Gaussian as in VAE. $f_\alpha : \mathbb{R}^d \to \mathbb{R}$ is the scalar-valued negative energy function and is parameterized by a small multi-layer perceptron (MLP) with parameters $\alpha$. $Z(\alpha) = \int \exp(f_\alpha(z))p_0(z)dz = \mathbb{E}_{p_0}[\exp(f_\alpha(z))]$ is the partition function.

The generation model $p_\beta(x|z)$ is a conditional autoregressive model,

$$p_\beta(x|z) = \prod_{t=1}^T p_\beta(x^{(t)}|x^{(1)}, ..., x^{(t-1)}, z) \tag{5}$$

which is parameterized by a one-layer LSTM with parameters $\beta$. Note that the latent vector $z$ controls every step of the autoregressive model.

## 3.3 JOINT DISTRIBUTION OF MOLECULE AND MOLECULAR PROPERTY

Given a molecule $x$, suppose $y$ is the chemical property of interest, such as QED or protein affinity binding. The property value can be computed from an input $x$ via open-sourced software RDKit (Landrum et al., 2013) or AutoDock-GPU (Santos-Martins et al., 2021). We assume that given $z$, $x$ and $y$ are conditionally independent.

$$p_\theta(x, y, z) = p_\alpha(z)p_\beta(x|z)p_\gamma(y|z), \tag{6}$$

where $p_\alpha(z)$ is the EBM prior, $p_\beta(x|z)$ is the generation model, and $p_\gamma(y|z)$ is the property regression model, and $\theta = (\alpha, \beta, \gamma)$. We use the model $p_\theta(x, y, z)$ to approximate the data distribution of $(x, y)$. See Appendix §A.1 for details.

The property regression model can be written as

$$p_\gamma(y|z) = \frac{1}{\sqrt{2\pi\sigma^2}} \exp\left(-\frac{1}{2\sigma^2}(y - s_\gamma(z))^2\right), \tag{7}$$

where $s_\gamma(z)$ is a small MLP, with parameters $\gamma$, predicting $y$ based on the latent $z$. The variance $\sigma^2$ is set as a constant or hyperparameter in our work.

## 3.4 LEARNING ALGORITHM

Suppose we observe training examples $\{(x_i, y_i), i = 1, ..., n\}$. The log-likelihood function is $L(\theta) = \sum_{i=1}^n \log p_\theta(x_i, y_i)$. The learning gradient can be calculated according to

$$\nabla_\theta \log p_\theta(x, y) = \mathbb{E}_{p_\theta(z|x,y)} [\nabla_\theta \log p_\theta(x, y, z)] \tag{8}$$

$$= \mathbb{E}_{p_\theta(z|x,y)} [\nabla_\theta(\log p_\alpha(z) + \log p_\beta(x|z) + \log p_\gamma(y|z))]. \tag{9}$$

For the prior model, $\nabla_\alpha \log p_\alpha(z) = \nabla_\alpha f_\alpha(z) - \mathbb{E}_{p_\alpha(z)}[\nabla_\alpha f_\alpha(z)]$. Thus the learning gradient given an example $(x, y)$ is

$$\delta_\alpha(x, y) = \nabla_\alpha \log p_\theta(x, y) = \mathbb{E}_{p_\theta(z|x,y)}[\nabla_\alpha f_\alpha(z)] - \mathbb{E}_{p_\alpha(z)}[\nabla_\alpha f_\alpha(z)]. \tag{10}$$

$\alpha$ is updated based on the difference between $z$ inferred from empirical observation $(x, y)$, and $z$ sampled from the current prior. For the generation model,

$$\delta_\beta(x, y) = \nabla_\beta \log p_\theta(x, y) = \mathbb{E}_{p_\theta(z|x,y)}[\nabla_\beta \log p_\beta(x|z)]. \tag{11}$$

Similarly, for the regression model,

$$\delta_\gamma(x, y) = \nabla_\gamma \log p_\theta(x, y) = \mathbb{E}_{p_\theta(z|x,y)}[\nabla_\gamma \log p_\gamma(y|z)]. \tag{12}$$

Estimating expectations in equations 17, 18, and 19 requires MCMC sampling of the prior model $p_\alpha(z)$ and the posterior distribution $p_\theta(z|x, y)$. We recruit Langevin dynamics (Neal, 2011). For a target distribution $\pi(z)$, the dynamics iterates

$$z_{k+1} = z_k + s\nabla_z \log \pi(z_k) + \sqrt{2s}\epsilon_k, \tag{13}$$

where $k$ indexes the time step of the Langevin dynamics, $s$ is a small step size, and $\epsilon_k \sim \mathcal{N}(0, I_d)$ is the Gaussian white noise. $\pi(z)$ can be either $p_\alpha(z)$ or $p_\theta(z|x, y)$. In either case, $\nabla_z \log \pi(z)$ can be efficiently computed by back-propagation. See Appendix §A.1 for more details.

---

**Algorithm 1:** Learning MolEBM.

**input** : Learning iterations $T$, learning rates for the prior, generation, and regression model $\{\eta_0, \eta_1, \eta_2\}$, initial parameters $\theta_0 = (\alpha_0, \beta_0, \gamma_0)$, observed examples $\{(x_i, y_i)\}_{i=1}^n$, batch size $m$, number of prior and posterior sampling steps $\{K_0, K_1\}$, and prior and posterior sampling step sizes $\{s_0, s_1\}$.

**output:** $\theta_T = (\alpha_T, \beta_T, \gamma_T)$.

**for** $t = 0 : T - 1$ **do**

    1. **Mini-batch**: Sample observed examples $\{(x_i, y_i)\}_{i=1}^m$.

    2. **Prior sampling**: For each $i$, sample $z_i^- \sim p_{\alpha_t}(z)$ using equation (13), where the target distribution $\pi(z) = p_{\alpha_t}(z)$, and $s = s_0, K = K_0$.

    3. **Posterior sampling**: For each $(x_i, y_i)$, sample $z_i^+ \sim p_{\theta_t}(z|x_i, y_i)$ using equation (13), where the target distribution $\pi(z) = p_{\theta_t}(z|x_i, y_i)$, and $s = s_1, K = K_1$.

    4. **Update prior model**: $\alpha_{t+1} = \alpha_t + \eta_0 \frac{1}{m} \sum_{i=1}^m [\nabla_\alpha f_{\alpha_t}(z_i^+) - \nabla_\alpha f_{\alpha_t}(z_i^-)]$.

    5. **Update generation model**: $\beta_{t+1} = \beta_t + \eta_1 \frac{1}{m} \sum_{i=1}^m \nabla_\beta \log p_{\beta_t}(x_i|z_i^+)$.

    6. **Update regression model**: $\gamma_{t+1} = \gamma_t + \eta_2 \frac{1}{m} \sum_{i=1}^m \nabla_\gamma \log p_{\gamma_t}(y_i|z_i^+)$.

---

### 3.5 SAMPLING WITH GRADUAL DISTRIBUTION SHIFTING

To tackle the single-objective optimization problem (eq. 1), one naive approach is to perform ancestral sampling with two steps, given some desirable property value $y^*$,

$$(1)\ z^* \sim p_\theta(z|y = y^*) \propto p_\alpha(z)p_\gamma(y = y^*|z) \quad (2)\ x^* \sim p_\beta(x|z = z^*), \tag{14}$$

where (1) is an application of Bayes' rule, with $p_\alpha(z)$ as the prior and $p_\gamma(y|z)$ as the likelihood.

Our model $p_\theta(x, y, z)$ is learned to capture the data distribution. In real-world settings, $y^*$ might not be within the support of the data distribution. Therefore, sampling following equation 14 does not work well since it involves extrapolating the learned distribution. We propose a method called *sampling with gradual distribution shifting* (SGDS) to address this issue. In particular, we find examples from the training data on the boundary of the distribution support by sorting them according to the values of $y$ and taking top-$k$ samples, $\{(x_i^{(\text{old})}, y_i^{(\text{old})})\}_{i=1}^k$, with high property values. We shift the support slightly by adding some small $\Delta_y$ to all $y$'s, and sample $x$ conditional on shifted $y$'s, following equation 14. Given the generated $x$'s, we compute their groundtruth $y$'s with RDKit or AutoDock-GPU. We then shift the learned model by finetuning it with the new data, $\{(x_i^{(\text{new})}, y_i^{(\text{new})})\}_{i=1}^k$, for a few steps (e.g., 10). This completes one iteration of distribution shift. The molecule with desired high property value is sampled after $T$ shifting iterations. We summarize the algorithm in Algorithm 2.

---

**Algorithm 2:** Sampling with Gradual Distribution Shifting (SGDS).

---

**input** : Shift iterations $T$, initial parameters $\theta_0 = (\alpha_0, \beta_0, \gamma_0)$, initial examples $\{(x_i^0, y_i^0)\}_{i=1}^k$ from the data distribution boundary, shift magnitude $\Delta_y$, PropertyComputeEngine = RDKit or AutoDock-GPU, LearningAlgorithm = Algorithm 1.

**output:** $\{(x_i^T, y_i^T)\}_{i=1}^k$.

**for** $t = 0 : T - 1$ **do**

    1. **Property shift**: For each $y_i^t$, $\tilde{y}_i^{t+1} = y_i^t + \Delta_y$.

    2. **Latent sampling**: For each $\tilde{y}_i^{t+1}$, sample $z_i^{t+1} \sim p_{\theta_t}(z|\tilde{y}_i^{t+1})$.

    3. **Molecule sampling**: For each $z_i^{t+1}$, sample $x_i^{t+1} \sim p_{\theta_t}(x|z_i^{t+1})$.

    4. **Property computation**: For each $x_i^{t+1}$, compute $y_i^{t+1} = \text{PropertyComputeEngine}(x_i^{t+1})$.

    5. **Distribution shift**: $\theta_{t+1} = \text{LearningAlgorithm}(\{(x_i^{t+1}, y_i^{t+1})\}_{i=1}^k, \theta_t)$.

---

For constrained optimization, in step 3, we only keep the sampled molecules that satisfy the given constraints.

### 3.6 MULTI-OBJECTIVE OPTIMIZATION

We next consider the multi-objective optimization problem. Suppose we optimize for a set of properties $\{y_j\}_{j=1}^m$, then we learn a property regression model for each property $y_j$,

$$p_{\gamma_j}(y_j|z) = \frac{1}{\sqrt{2\pi\sigma_j^2}} \exp\left(-\frac{1}{2\sigma_j^2}(y_j - s_{\gamma_j}(z))^2\right), \tag{15}$$

where each $s_{\gamma_j}$ is a small MLP with parameters $\gamma_j$. Then the joint distribution is,

$$p_\theta(x, z, y_1, ..., y_m) = p_\alpha(z)p_\beta(x|z) \prod_{j=1}^m p_{\gamma_j}(y_i|z). \tag{16}$$

Under our framework, the learning algorithm and the sampling algorithm for the single-objective problem can be straightforwardly extended to the multi-objective setting. In both settings, the same types of properties are provided in both the initial training stage and the distribution shifting stage.

## 4 EXPERIMENTS

To demonstrate the effectiveness of our proposed model, we compare our model with previous SOTA methods for unconditional generation (§4.2) and molecule design including single-objective optimization (§4.3) and multi-objective optimization (§4.4). In molecule design experiments, we consider both non-biological and biological properties.

### 4.1 EXPERIMENTAL SETUP

**Datasets.** For the unconditional generation task, we report results on ZINC (Irwin et al., 2012) and MOSES datasets (Polykovskiy et al., 2020). ZINC consists of around $250k$ molecules, and MOSES comprises around 2 million molecules. For optimization tasks, we conduct experiments with ZINC.

**Training Details.** There are three modules in our model, the top-down generation model $p_\beta(x|z)$, the EBM prior $p_\alpha(z)$, and the regression model $p_\gamma(y|z)$. $p_\beta(x|z)$ is parameterized by a single-layer LSTM with 1024 hidden units. The dimension of latent vector $z$ is 100. $p_\alpha(z)$ is a 3-layer MLP with 100 input dimension and 200 hidden units. The property regression model $p_\gamma(y|z)$ is a 3-layer MLP with 100 input dimension and 100 hidden units. In multi-objective optimization settings, we recruit one MLP for each property and these MLPs have the same architecture as mentioned above. It is worth mentioning that compared to most previous models, our model is characterized by its simplicity without adding inference networks, RL-related modules, and graph neural networks. Adam (Kingma & Ba, 2015) optimizer is used to train our models with learning rates 0.0001 for EBM and 0.001 for the rest. We train our models for 30 epochs. $10,000$ (non-biological) and $2,000$

(biological) boundary examples are selected for distribution shifting. The numbers of shift iterations are 50 (non-biological single-objective), 20 (non-biological single-objective) and 10 (multi-objective). Within each SGDS iteration, the model is finetuned with only 10 parameter updates. All experiments are conducted on Nvidia Titan XP GPU. See more details in Appendix §A.6.

## 4.2 UNCONDITIONAL GENERATION

Three types of encoding systems are used to encode molecules in prior work: SMILES (Weininger, 1988), SELFIES (Krenn et al., 2020), and graph. SMILES and SELFIES linearize a molecule graph into a string of characters. Most previous models using SMILES struggle to generate molecules with high validity, which is the percentage of molecules that satisfy the chemical valency rule. Thus graph representations become popular since explicit valency constraints can be imposed. However, perfect validity in this approach does not imply that the model captures the chemical rules since it is achieved with external constraints. Recently, SELFIES is developed where every SELFIES string corresponds to a valid molecule due to the nature of the encoding system. Thus, validity for generated SMILES strings is a good indicator on how well a learned model captures the basic chemical rules implicitly. Besides validity, we also compare models on uniqueness (the percentage of unique molecules in all generated samples) and novelty (the percentage of generated molecules that are not in the training set).

Following previous work, we randomly sample $10k$ molecules for ZINC and $30k$ for MOSES, and compare on the three aforementioned metrics. Generations results on ZINC and MOSES are shown in Table 1 and Table 2 respectively. `str-smi` denotes string-based SMILES representations and `str-sfi` denotes string-based SELFIES representations. In Table 1, we show generation results for both SMILES and SELFIES. SMILES does not have a validity constraint during generation. However, our model, MolEBM, still achieves $95.5\%$ validity, which outperforms other SMILES-based methods and is also comparable to those with valency check. This result demonstrates that our model can capture those valency rules effectively and implicitly (also see Appendix §A.4.). Samples from our model also achieve perfect uniqueness and novelty.

| Model | Representation | Validity | Novelty | Uniqueness |
|---|---|---|---|---|
| JT-VAE (Jin et al., 2018) | Graph | 1.000* | 1.000 | 1.000 |
| GCPN (You et al., 2018) | Graph | 1.000* | 1.000 | 1.000 |
| GraphNVP (Madhawa et al., 2019) | Graph | 0.426 | 1.000 | 0.948 |
| GraphAF (Shi et al., 2020) | Graph | 1.000* | 1.000 | 0.991 |
| GraphDF (Luo et al., 2021) | Graph | 1.000* | 1.000 | 1.000 |
| ChemVAE (Gómez-Bombarelli et al., 2018) | `str-smi` | 0.170 | 0.980 | 0.310 |
| GrammarVAE (Kusner et al., 2017) | `str-smi` | 0.310 | 1.000 | 0.108 |
| SDVAE (Dai et al., 2018) | `str-smi` | 0.435 | - | - |
| **MolEBM** | `str-smi` | 0.955 | 1.000 | 1.000 |
| **MolEBM** | `str-sfi` | 1.000 | 1.000 | 1.000 |

Table 1: Unconditional generation on ZINC. * denotes valency check.

| Model | Representation | Validity | Novelty | Uniqueness |
|---|---|---|---|---|
| JT-VAE (Jin et al., 2018) | Graph | 1.000* | 0.914 | 1.000 |
| GraphAF (Shi et al., 2020) | Graph | 1.000* | 1.000 | 0.991 |
| GraphDF (Luo et al., 2021) | Graph | 1.000* | 1.000 | 1.000 |
| LIMO (Eckmann et al., 2022) | `str-sfi` | 1.000 | 1.000 | 0.976 |
| **MolEBM** | `str-sfi` | 1.000 | 1.000 | 1.000 |

Table 2: Unconditional generation on MOSES. * denotes valency check. Results obtained from (Polykovskiy et al., 2020; Eckmann et al., 2022).

## 4.3 SINGLE-OBJECTIVE OPTIMIZATION

**Non-Biological Property Optimization.** For non-biological properties, we are interested in Penalized logP and QED, both of which can be calculated by RDKit (Landrum et al., 2013). Since we know the Penalized logP scores have a positive relationship with the length of the molecules, we

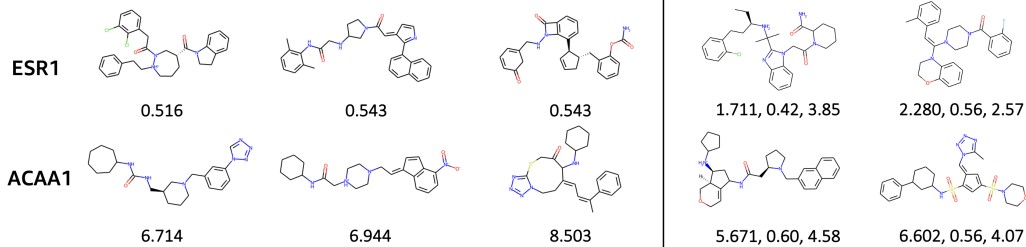

Figure 2: Generated molecules with high binding affinities. Left: Top-3 single-objective optimized molecules. The number denotes $\mathbf{K_D}$ in nmol/L. Right: Top-2 multi-objective optimized molecules. Numbers denote $\mathbf{K_D}$ in nmol/L, QED and SA respectively.

maximize Penalized logP either with or without maximum length limit. Following (Eckmann et al., 2022), the maximum length is set to be the maximum length of molecules in ZINC using SELFIES. From Table 9, we can see that with length limit, MolEBM outperforms previous methods by a large margin. Maximizing penalized logP without length limit leads to the highest Penalized logP, $158.0$. We also achieve the highest QED with/without length limit. These observations demonstrate the effectiveness of our method. We also illustrate our distribution shifting method in Appendix §A.3.

| Method | LL | Penalized logP ($\uparrow$) | | | QED ($\uparrow$) | | |
|---|---|---|---|---|---|---|---|
| | | 1st | 2rd | 3rd | 1st | 2rd | 3rd |
| JT-VAE | ✗ | 5.30 | 4.93 | 4.49 | 0.925 | 0.911 | 0.910 |
| GCPN | ✓ | 7.98 | 7.85 | 7.80 | **0.948** | 0.947 | 0.946 |
| MolDQN | ✓ | 11.8 | 11.8 | 11.8 | **0.948** | 0.943 | 0.943 |
| MARS | ✗ | 45.0 | 44.3 | 43.8 | **0.948** | **0.948** | **0.948** |
| GraphDF | ✗ | 13.7 | 13.2 | 13.2 | **0.948** | **0.948** | **0.948** |
| LIMO | ✓ | 10.5 | 9.69 | 9.60 | 0.947 | 0.946 | 0.945 |
| **MolEBM** | ✓ | **26.4** | **25.1** | **24.4** | **0.948** | **0.948** | **0.948** |
| **MolEBM** | ✗ | **158.0** | **157.8** | **157.5** | **0.948** | **0.948** | **0.948** |

Table 3: Non-biological single-objective optimization. Report top-3 highest scores found by each model. LL (Length Limit) denotes whether the model has the limit of maximum length. Baseline results obtained from (Eckmann et al., 2022; You et al., 2018; Luo et al., 2021; Xie et al., 2021).

**Biological Property Optimization.** ESR1 and ACAA1 are two human proteins. We aim to design ligands (molecules) that have the maximum binding affinities towards those target proteins. ESR1 is well-studied, which has many existing binders, while ACAA1 does not. However, we did not use any binder-related information in the design process. Binding affinity is measured by the estimated dissociation constants $\mathbf{K_D}$, which can be computed with AutoDock-GPU (Santos-Martins et al., 2021) given a molecule. Large binding affinities corresponds to small $\mathbf{K_D}$. That is, we aim to minimize $\mathbf{K_D}$. Table 4 shows that our model outperforms previous methods on both ESR1 and ACAA1 binding affinity maximization tasks. Producing those ligands with high binding affinity plays a vital role in the early stage of drug discovery.

| Method | ESR1 $\mathbf{K_D}$ ($\downarrow$) | | | ACAA1 $\mathbf{K_D}$ ($\downarrow$) | | |
|---|---|---|---|---|---|---|
| | 1st | 2rd | 3rd | 1st | 2rd | 3rd |
| GCPN | 6.4 | 6.6 | 8.5 | 75 | 83 | 84 |
| MolDQN | 373 | 588 | 1062 | 240 | 337 | 608 |
| MARS | 25 | 47 | 51 | 370 | 520 | 590 |
| GraphDF | 17 | 64 | 69 | 163 | 203 | 236 |
| LIMO | 0.72 | 0.89 | 1.4 | 37 | 37 | 41 |
| **MolEBM** | **0.52** | **0.54** | **0.54** | **6.71** | **6.94** | **8.50** |

Table 4: Biological single-objective optimization. Report top-3 lowest $\mathbf{K_D}$ (in nanomoles/liter) found by each model. Baseline results obtained from (Eckmann et al., 2022).

### 4.4 MULTI-OBJECTIVE OPTIMIZATION

We next consider optimizing binding affinity, QED and SAS simultaneously for multi-objective optimization. Following Eckmann et al. (2022), we exclude molecules with abnormal behaviors [1] in the generation process to make sure the learned distribution shifts towards a desirable region in terms of pharmacologic and synthetic properties.

Table 5 shows our multi-objective binding affinity maximization results comparing to LIMO (Eckmann et al., 2022) and GCPN (You et al., 2018). From the results, we can see that MolEBM is able to find the ligands with desired properties while keeping the pharmacologic structures. For ESR1, we have two existing binders on the market, Tamoxifen and Raloxifene. Our designed ligands have similar QED and SA, with rather low $K_D$. Compared to existing methods, MolEBM obtains better results in overall adjustments. For ACAA1, we do not have any existing binders. Compared with prior SOTA methods, our optimized ligands outperform those by a large margin. When comparing ACAA1 multi-objective setting with its corresponding single-objective one, we find multi-objective results even outperforms the single-objective one, which may be counter-intuitive since multi-objective optimization is assumed to be a harder task than single-objective optimization. However, our current results indicate that with proper prior knowledge (e.g. multiple objectives are positively aligned), multi-objective settings could be more plausible in *de novo* design, and those complicated prior knowledge can indeed be captured by our expressive latent space EBM. While we still need domain expertise to determine the effectiveness of those ligands discovered by MolEBM, we believe our model could shed light on the optimization procedure in the early drug discovery.

| Ligand | ESR1 | | | ACAA1 | | |
|---|---|---|---|---|---|---|
| | $K_D$ ($\downarrow$) | QED ($\uparrow$) | SA ($\downarrow$) | $K_D$ ($\downarrow$) | QED ($\uparrow$) | SA ($\downarrow$) |
| GCPN $1^{st}$ | 810 | 0.43 | 4.2 | 8500 | 0.69 | 4.2 |
| GCPN $2^{nd}$ | $2.7 \times 10^4$ | 0.80 | 3.7 | 8500 | 0.54 | 4.3 |
| LIMO $1^{st}$ | 4.6 | 0.43 | 4.8 | 28 | 0.57 | 5.5 |
| LIMO $2^{nd}$ | 2.8 | 0.64 | 4.9 | 31 | 0.44 | 4.9 |
| Tamoxifen | 87 | 0.45 | 2.0 | – | – | – |
| Raloxifene | $7.9 \times 10^6$ | 0.32 | 2.4 | – | – | – |
| **MolEBM** $1^{st}$ | **1.71** | 0.42 | 3.85 | **5.67** | 0.60 | 4.58 |
| **MolEBM** $2^{nd}$ | **2.28** | 0.56 | 2.56 | **6.60** | 0.56 | 4.07 |

Table 5: Muli-Objective Binding Affinity Maximization for both ESR1 and ACAA1. Report Top-2 average scores related to $K_D$ (in nmol/L), QED and SA. Baseline results obtained from (Eckmann et al., 2022).

Figure 2 shows generated molecules with high binding affinities. See Appendix for more examples.

## 5 CONCLUSION AND DISCUSSION

We propose a deep generative model, MolEBM, which models the joint distribution of molecules and molecular properties. It assumes an energy-based prior for a low-dimensional continuous latent space, which effectively captures data regularities of the discrete molecule data. We then design a distribution shifting method (SGDS) to shift the learned distribution to a region with high property values. Molecule design can then be achieved by conditional sampling. Our experiments demonstrate that our method outperforms previous SOTA methods by a significant margin.

A limiting factor is that the sampling with gradual distribution shifting (SGDS) requires that the properties of interest can be easily computed. In our future work, we shall explore semi-supervised learning methods using our model where the properties are available only for a small sample of molecules.

Another limitation is that, for our optimization method, good property scores may not translate to useful molecules for drug design. This problem may be partially addressed by multi-objective optimization as studied in this paper. In our work, we focus on optimizing given objectives. Designing good objectives or metrics is an equally or perhaps even more important problem that deserves careful investigation.

---

[1]In particular, we exclude molecules with QED ($\uparrow$) smaller than 0.4, SA ($\downarrow$) larger than 5.5, and too small (less than 5 atoms) or too large (more than 6 atoms) chemical rings.

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

## A  APPENDIX

### A.1  DETAILS ABOUT MODEL AND LEARNING

Our model is of the form $p_\alpha(z)p_\beta(x|z)p_\gamma(y|z)$. The marginal distribution of $(x, y)$ is

$$p_\theta(x, y) = \int p_\theta(x, y, z)dz = \int p_\alpha(z)p_\beta(x|z)p_\gamma(y|z)dz.$$

We use $p_\theta(x, y)$ to approximate the data distribution of $(x, y)$.

For the data distribution of $(x, y)$, $y$ is a deterministic function of $x$. However, a machine learning method usually cannot learn the deterministic function exactly. Instead, we can only learn a probabilistic $p_\theta(y|z)$. Our model $p_\theta(x, y)$ seeks to approximate the data distribution $p(x, y)$ by maximum likelihood. A learnable and flexible prior model $p_\alpha(z)$ helps to make the approximate more accurate than a fixed prior model such as that in VAE.

Let the training data be $\{(x_i, y_i), i = 1, ..., n\}$. The log-likelihood function is $L(\theta) = \sum_{i=1}^{n} \log p_\theta(x_i, y_i)$. The learning gradient is $L'(\theta) = \sum_{i=1}^{n} \nabla_\theta \log p_\theta(x_i, y_i)$. In the following, we provide details for calculating $\nabla_\theta \log p_\theta(x, y)$ for a single generic training example $(x, y)$ (where we drop the subscript $_i$ for notation simplicity).

$$
\begin{aligned}
\nabla_\theta \log p_\theta(x, y) &= \frac{1}{p_\theta(x, y)} \nabla_\theta p_\theta(x, y) \\
&= \frac{1}{p_\theta(x, y)} \int \nabla_\theta p_\theta(x, y, z)dz \\
&= \frac{1}{p_\theta(x, y)} \int p_\theta(x, y, z) \nabla_\theta \log p_\theta(x, y, z)dz \\
&= \int \frac{p_\theta(x, y, z)}{p_\theta(x, y)} \nabla_\theta \log p_\theta(x, y, z)dz \\
&= \int p_\theta(z \mid x, y) \nabla_\theta \log p_\theta(x, y, z)dz \\
&= \mathbb{E}_{p_\theta(z|x,y)} [\nabla_\theta \log p_\theta(x, y, z)] \\
&= \mathbb{E}_{p_\theta(z|x,y)} [\nabla_\theta (\log p_\alpha(z) + \log p_\beta(x|z) + \log p_\gamma(y|z))].
\end{aligned}
$$

For the prior model,

$$
\begin{aligned}
\nabla_\alpha \log p_\alpha(z) &= \nabla_\alpha f_\alpha(z) - \nabla_\alpha \log Z(\alpha) \\
&= \nabla_\alpha f_\alpha(z) - \frac{1}{Z(\alpha)} \nabla_\alpha Z(\alpha) \\
&= \nabla_\alpha f_\alpha(z) - \frac{1}{Z(\alpha)} \int \nabla_\alpha \exp(f_\alpha(z))p_0(z)dz \\
&= \nabla_\alpha f_\alpha(z) - \int \nabla_\alpha f_\alpha(z) \frac{1}{Z(\alpha)} \exp(f_\alpha(z))p_0(z)dz \\
&= \nabla_\alpha f_\alpha(z) - \mathbb{E}_{p_\alpha(z)} [\nabla_\alpha f_\alpha(z)].
\end{aligned}
$$

Thus the learning gradient for $\alpha$ given an example $(x, y)$ is

$$\delta_\alpha(x, y) = \nabla_\alpha \log p_\theta(x, y) = \mathbb{E}_{p_\theta(z|x,y)}[\nabla_\alpha f_\alpha(z)] - \mathbb{E}_{p_\alpha(z)}[\nabla_\alpha f_\alpha(z)]. \tag{17}$$

The above equation has an empirical Bayes nature. $p_\theta(z|x, y)$ is based on the empirical observation $(x, y)$, while $p_\alpha$ is the prior model. For the generation model,

$$\delta_\beta(x, y) = \nabla_\beta \log p_\theta(x, y) = \mathbb{E}_{p_\theta(z|x,y)}[\nabla_\beta \log p_\beta(x|z)]. \tag{18}$$

Similarly, for the regression model,

$$\delta_\gamma(x, y) = \nabla_\gamma \log p_\theta(x, y) = \mathbb{E}_{p_\theta(z|x,y)}[\nabla_\gamma \log p_\gamma(y|z)]. \tag{19}$$

Estimating expectations in the above equations requires Monte Carlo sampling of the prior model $p_\alpha(z)$ and the posterior distribution $p_\theta(z|x, y)$. If we can draw fair samples from the two distributions, and use these Monte Carlo samples to approximate the expectations, the the gradient ascent algorithm based on the Monte Carlo samples is the stochastic gradient ascent algorithm or the stochastic approximation algorithm of Robbins and Monro (Robbins & Monro, 1951), who established the convergence of such an algorithm to a local maximum of the log-likelihood.

For MCMC sampling using Langevin dynamics, the finite step or short run Langevin dynamics may cause bias in Monte Carlo sampling. The bias was analyzed in Pang et al. (2020). The resulting algorithm is an approximate maximum likelihood learning algorithm.

## A.2 ABLATION STUDIES

We conduct ablations on the key components of our method: 1) EBM Prior (EBM vs. standard Gaussian $\mathcal{N}(\mathbf{0}, \mathbf{I})$), (2) SGDS (shift distributions with SGDS vs. no shift), and (3) joint training (joint training of molecule and molecular property distribution vs. train molecule with LSEBM and learn property regression model in a second step). The ablation results are displayed in Table 6. It is clear that all the proposed components contribute significantly to the good performance of our method.

| EBM Prior | SGDS | Joint Training | Penalized logP | | |
|:---:|:---:|:---:|:---:|:---:|:---:|
| | | | 1st | 2nd | 3rd |
| ✗ | ✓ | ✓ | 13.81 | 13.78 | 13.75 |
| ✓ | ✗ | ✓ | 12.27 | 12.02 | 11.93 |
| ✓ | ✓ | ✗ | 12.79 | 12.70 | 12.41 |
| ✓ | ✓ | ✓ | **26.37** | **25.05** | **24.38** |

Table 6: Ablation studies.

## A.3 ILLUSTRATION OF SAMPLING WITH GRADUAL DISTRIBUTION SHIFTING (SGDS)

Figure 3 shows property score densities of sampled molecules from our MolEBM in the distribution shifting process. We can see the model distribution is gradually shifting towards the region supported by molecules with high property values.

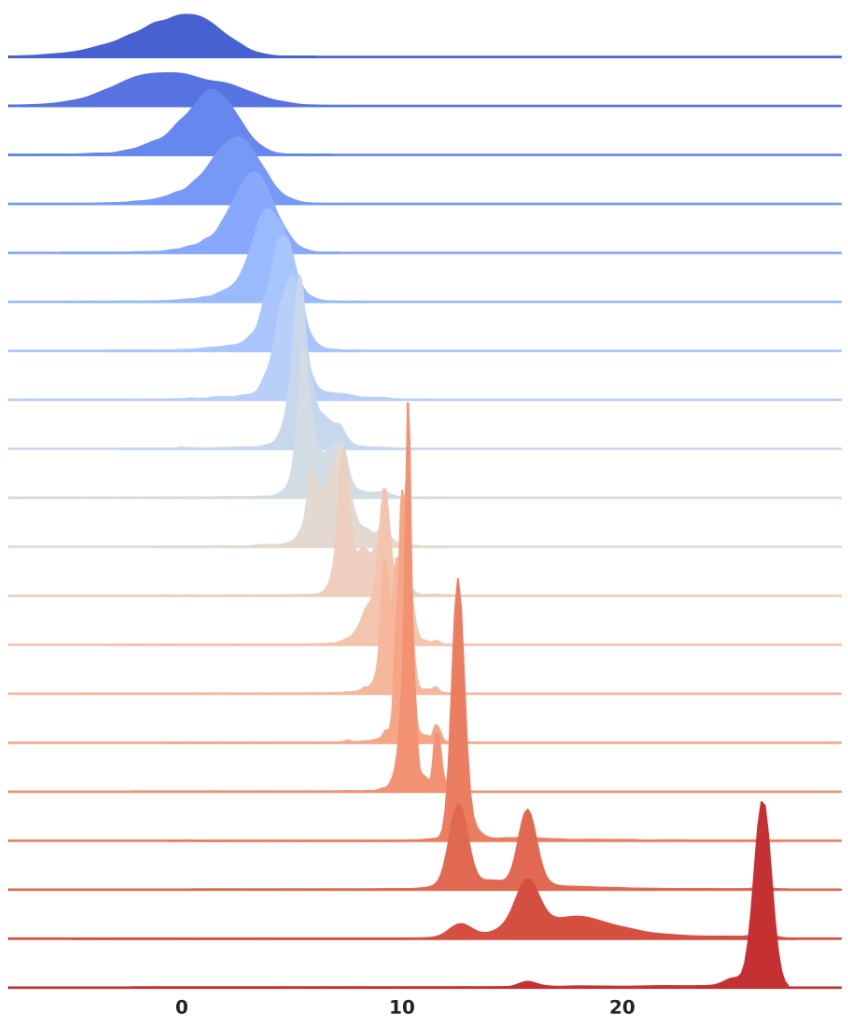

Figure 3: Illustration of SGDS in a single-objective penalized logP optimization experiment.

## A.4 UNCONDITIONAL GENERATION

For unconditional generation task, we use uniqueness, novelty and validity to compare our generation results with existing methods. Meanwhile, for SELFIES-based ZINC dataset, we split this dataset into train split ($240k$) and test split ($10k$ samples). Here, we randomly sample $10k$ molecules from the learned latent EBM and calculate their logP, QED and SA scores using RDKit. We compare these property densities with molecule property densities in test split. The results are shown in Figure 4. We can see that the marginal distributions (property score densities) of the learned model match those of the data quite well, implying that our model indeed captures the chemical rules implicitly.

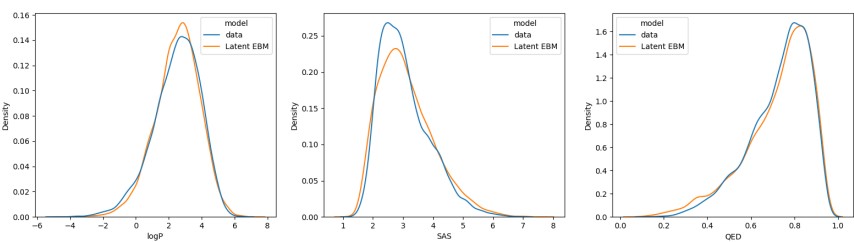

Figure 4: Property score distributions.

## A.5 Generation with Different Length of Markov Chain

In our experiments, we use short-run MCMC (i.e. $s = 20$) in Equation 13 for all experiments. From Figure 5, we can see with the increasing length of Markov chains, the molecules change accordingly, showing that Markov chain doesn't get stuck in the local mode.

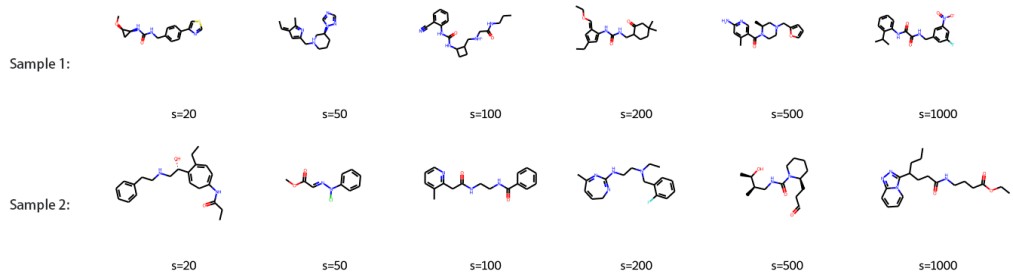

Figure 5: Sampled molecules with the different length of Markov chain.

## A.6 Training Time

The joint training of MolEBM takes 4 hours with 30 iterations on a single Nvidia Titan XP GPU with batch size 2048. For non-biological single-objective property optimization, it takes around 0.5 hours to do 50 distribution shifting (SGDS) iterations. For biological binding affinity maximization, the optimization time is mainly dependent on the number of queries of AutoDock-GPU. We do 20 and 10 SGDS iterations for the single-objective and multi-objective tasks, respectively, which cost 10 hours and 5 hours. For biological property optimization tasks, we use two Nvidia Titan XP GPUs, one for running our code and another one for running AutoDock-GPU. We have added a table to compare with previous methods.

| Model | Penalized-logP/QED |
|---|---|
| JT-VAE | 24 |
| GCPN | 8 |
| MolDQN | 24 |
| GraphDF | 8 |
| Mars | 12 |
| LIMO | 1 |
| MolEBM | 4.5 |

Table 7: Comparison of molecule generation time in (hrs). Results obtained from (Eckmann et al., 2022).

Even if we intensively use MCMC sampling-based methods, our training speed is affordable comparing to existing methods. That's due to our designed latent space EBM is low-dimensional (i.e. $\dim(z)=100$) and we use short-run MCMC (i.e. with fixed iteration steps $s = 20$) in our experiments. The sampling results with respect to the length of Markov chain is discussed in previous section.

## A.7 Optimization with Constraints

In unconditional generation task, even with SMILES representation, our model can capture the valency constraints in chemical space. This idea can further be extended to constrained optimizations, and those constraints in the chemical space can be directly imposed by keeping the sampled molecules that satisfy the constraints during the SGDS molecule sampling step in Algorithm 2. We include logP targeting and similarity-constrained optimization experiments in the following.

For example, in logP targeting experiments, during each iteration, we only keep the molecules within the target logP range; in similarity-constrained experiments, we use those molecules which satisfy similarity constraints to update the model. By iteratively updating the model with those selected molecules, we shift the joint distribution towards the region that satisfies the constraints.

### A.7.1 logP Targeting

Comparing to previous methods, MolEBM is able to get competitive diversity scores with significantly better success rate in both ranges. That's because after SGDS, our model is shifted towards the region that is supported by molecules satisfying the logP constraints. Due to the flexibility of our EBM prior, MolEBM achieves rather high diversity scores while keeping most of the sampled molecules within the logP range.

| Method | $-2.5 \leq \text{logP} \leq -2$ | | $5 \leq \text{logP} \leq 5.5$ | |
|---|---|---|---|---|
| | Success | Diversity | Success | Diversity |
| ZINC | 0.4% | 0.919 | 1.3% | 0.901 |
| JT-VAE | 11.3% | 0.846 | 7.6% | 0.907 |
| ORGAN | 0 | − | 0.2% | **0.909** |
| GCPN | 85.5% | 0.392 | 54.7% | 0.855 |
| LIMO | 10.4% | **0.914** | − | − |
| **MolEBM** | **86.0**% | 0.874 | **62.2**% | 0.858 |

Table 8: logP targeting to a certain range (Eckmann et al., 2022; You et al., 2018; Luo et al., 2021; Xie et al., 2021).

### A.7.2 Similarity-constrained Optimization

Following previous procedures in JT-VAE (Jin et al., 2018), we select 800 molecules with the lowest penalized-logP scores in ZINC250k dataset. This experiment aims to generate novel molecules with high penalized-log while similarity to the target molecules. We first randomly select one molecule as the target, and then optimize the p-logP. For each SGDS step, we only keep the molecules that have similarity score greater than minimum value $\delta$.

| $\delta$ | GCPN | | GraphDF | | LIMO | | MolEBM | |
|---|---|---|---|---|---|---|---|---|
| | Improv. | % Succ. | Improv. | % Succ. | Improv. | % Succ. | Improv. | % Succ. |
| 0.0 | $4.2 \pm 1.3$ | **100** | $5.9 \pm 2.0$ | **100** | $10.1 \pm 2.3$ | **100** | $\mathbf{19.11 \pm 2.12}$ | **100** |
| 0.2 | $4.1 \pm 1.2$ | **100** | $5.6 \pm 1.7$ | **100** | $5.8 \pm 2.6$ | 99.0 | $\mathbf{7.41 \pm 1.89}$ | **100** |
| 0.4 | $2.5 \pm 1.3$ | **100** | $\mathbf{4.1 \pm 1.4}$ | **100** | $3.6 \pm 2.3$ | 93.7 | $3.80 \pm 1.44$ | 97.5 |

Table 9: Similarity-constrained optimization results. Baseline results obtained from (Eckmann et al., 2022; Luo et al., 2021).

### A.8 Additional Experiments on GuacaMol Benchmarks

We further evaluate our MolEBM on several distribution learning and goal-directed optimization tasks in GuacaMol benchmark (Brown et al., 2019).

To be specific, for distribution learning, we use the validity, uniqueness and novelty to evaluate our model. The results are shown in Table 10. Comparing to existing methods, our MolEBM trained using SELFIES representations achieves highest scores among all three tasks.

For goal-directed benchmarks, we select five multiple property optimization tasks (MPO). Due to the time limit, we did not have time to tune our code for the new experiments. We expect to obtain improved results with fine-tuning. However, our MolEBM is still able to achieve comparable results to the strong baseline as Graph GA.

Again, GuacaMol is an extremely useful benchmark since it provides the normalized scoring functions which are weighed sum of multiple diverse properties of interest. The normalized scoring functions enable the fast convergence of MolEBM. With GuacaMol, we can also investigate the design choice between single property regression network to predict the pre-defined scores and multiple property regression networks to predict original chemical properties individually. We leave this question for future studies.

### A.8.1 Distribution-Learning Benchmarks

| Benchmark | AAE | Graph MCTS | Random Sampler | SMILES LSTM | VAE | **MolEBM** |
|---|---|---|---|---|---|---|
| Validity | 0.822 | **1.000** | **1.000** | **1.000** | 0.959 | **1.000** |
| Uniqueness | **1.000** | **1.000** | 0.997 | **1.000** | 0.999 | **1.000** |
| Novelty | 0.998 | 0.994 | 0.000 | 0.912 | 0.971 | **0.999** |

Table 10: Distribution learning results on GuacaMol benchmarks (Brown et al., 2019).

### A.8.2 Goal-directed Benchmarks

| Benchmark | Best of Dataset | SMILES GA | Graph MCTS | Graph GA | SMILES LSTM | **MolEBM** |
|---|---|---|---|---|---|---|
| Osimertinib MPO | 0.839 | 0.886 | 0.784 | **0.953** | 0.907 | *0.933* |
| Fexofenadine MPO | 0.817 | 0.931 | 0.695 | **0.998** | 0.959 | *0.971* |
| Ranolazine MPO | 0.792 | 0.881 | 0.616 | *0.920* | 0.855 | **0.924** |
| Sitagliptin MPO | 0.509 | 0.689 | 0.458 | **0.891** | 0.545 | *0.829* |

Table 11: Goal-directed optimization results on GuacaMol benchmarks (Brown et al., 2019). Top-2 results are highlighted as **bold** and *italic* respectively.

### A.9 Generated Samples

Figure 6 and figure 7 show generated molecules with high binding affinities towards ESR1 and ACAA1 respectively in single-objective property design experiments.

Figure 8 and Figure 9 show generated molecules with high binding affinities towards ESR1 and ACAA1 respectively in multi-objective property design.

Comparing to the previous state-of-the-art methods, our MolEBM is able to produce more high quality molecules than top-3 molecules because after sampling with gradual distribution shifting (SGDS), MolEBM locates at the area supported by molecules with high binding affinities.

In single-objective design, we find that few generated molecules may be of less practical use due to undesired properties (e.g. the first one in Figure 7 has a large circle). This observation is in accordance with (Eckmann et al., 2022), which is the case where the single-objective optimization is not sufficient. Thus we need multi-objective binding affinities design settings because in contrast to non-biological properties, binding affinities are hard to compute and optimize.

In multi-objective design settings, we find that those issues mentioned above can be partially addressed by optimizing binding affinities, QED and SA at the same time.

Meanwhile, compared to previous generative model based methods, we use Langevin dynamics to infer the posterior distribution $p(z|x, y_1, \ldots, y_n)$ without bothering to design different encoders when facing different combination of properties.

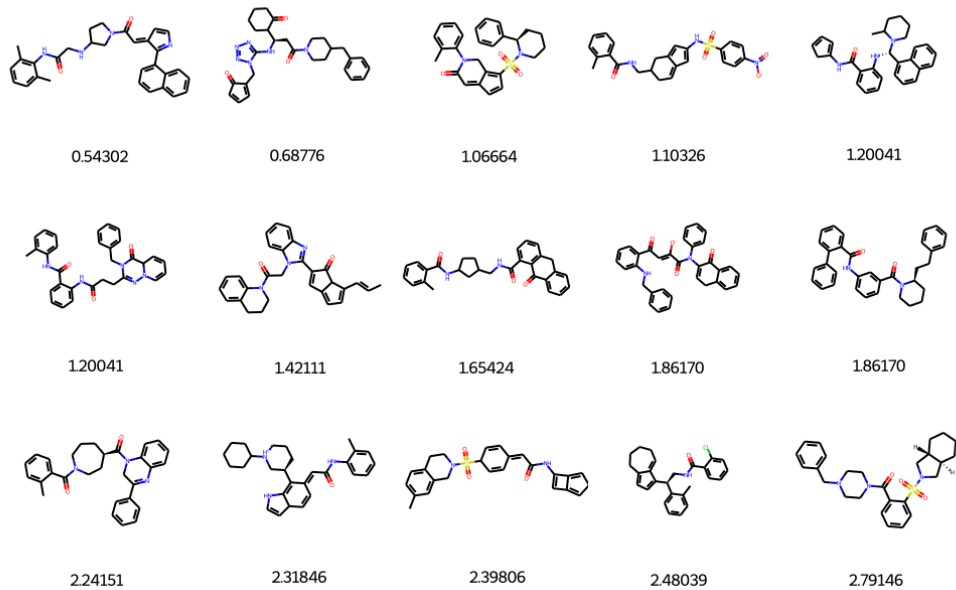

Figure 6: Generated molecules in **singe-objective** esr1 binding affinity maximization experiments with corresponding $\mathbf{K_D}(\downarrow)$ in nmol/L.

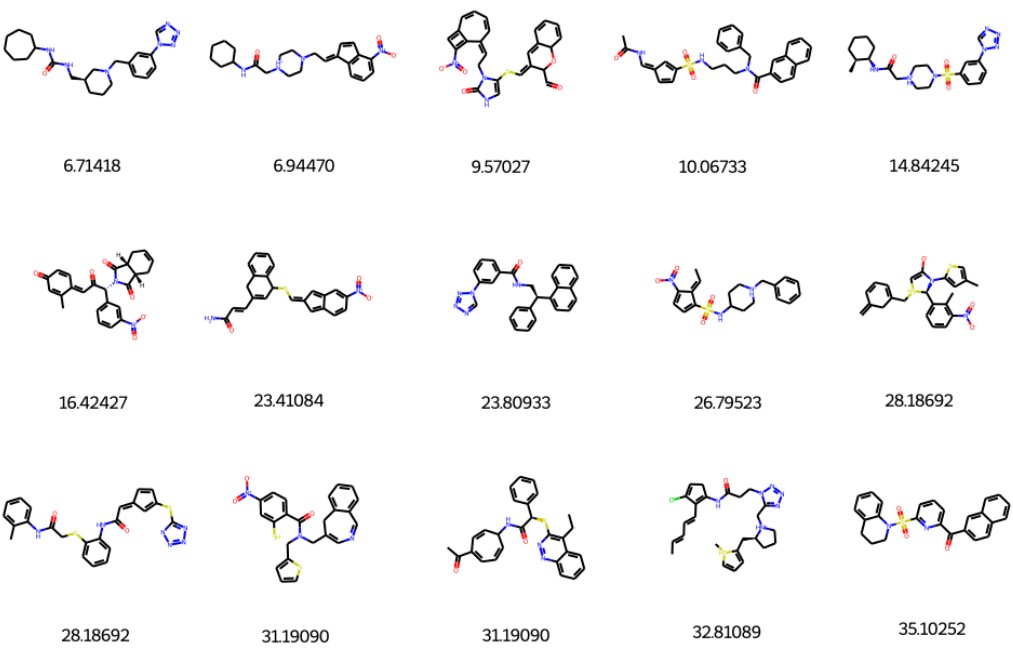

Figure 7: Generated molecules in **singe-objective** acaa1 binding affinity maximization experiments with corresponding $\mathbf{K_D}(\downarrow)$ in nmol/L.

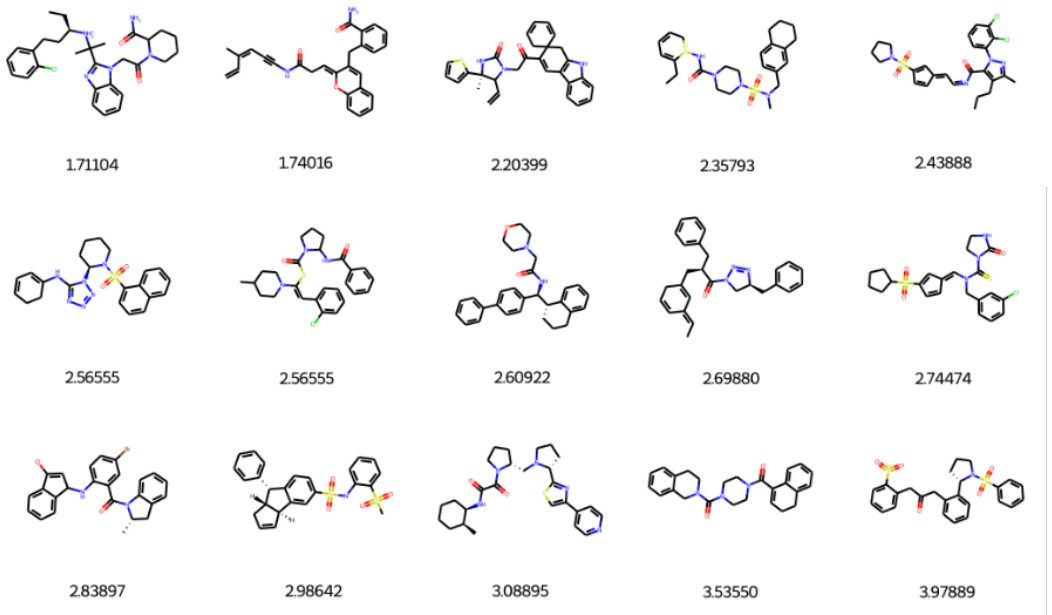

Figure 8: Generated molecules in **multi-objective** esr1 binding affinity maximization experiments with corresponding $\mathbf{K_D}(\downarrow)$ in nmol/L.

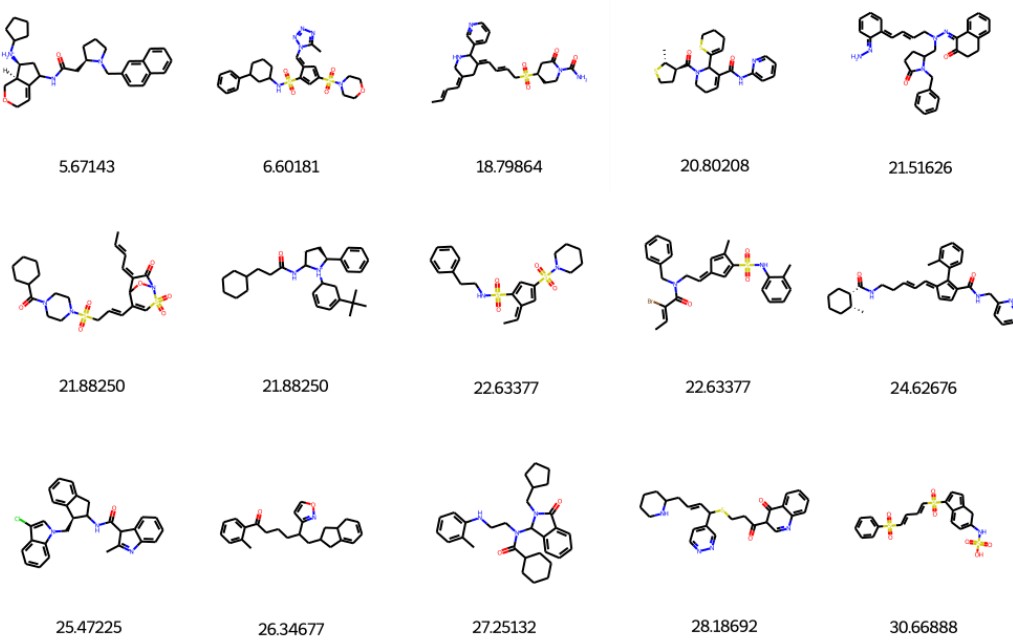

Figure 9: Generated molecules in **multi-objective** acaa1 binding affinity maximization experiments with corresponding $\mathbf{K_D}(\downarrow)$ in nmol/L.

## A.10 TOP-3 MOLECULES IN p-LOGP AND QED OPTIMIZATION

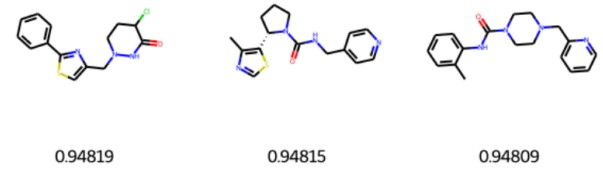

Figure 10: Top-3 molecules in single-objective QED maximization.

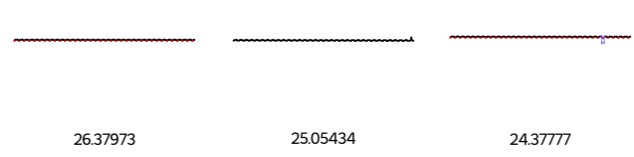

Figure 11: Top-3 molecules in single-objective p-logP maximization.

## A.11 REPRODUCIBILITY

Our code and saved checkpoints can be found here [2].

---

[2]https://drive.google.com/drive/folders/1UQcXrLWo20wuBocCIEIq7RRm1p2-bb8H?usp=sharing

