# OpenReview forum: "MolEBM: Molecule Generation and Design by Latent Space Energy-Based Modeling"
_ICLR.cc/2023/Conference — Submitted to ICLR 2023_

### Official Review · Reviewer_vKUH · 2022-10-20

**Confidence:** 4
**Correctness:** 2
**Technical Novelty And Significance:** 2
**Empirical Novelty And Significance:** 2
**Recommendation:** 5

**Clarity, Quality, Novelty And Reproducibility:**

Code and trained models are available to download.

Regarding clarity -
- what does ‘expressive’ latent space mean? I think it means that points which are close together in latent space correspond to molecules with similar properties, but this should be clarified.
- 3.3 is hard to make sense of. We are told first that y is a deterministic function of x, and then that x and y are conditionally independent given z. This can only be true if y is a deterministic function of z, but in equation (7) we’re told that y has Gaussian distribution given z.  Instead, one could just say that the model fits a distribution of the form given in equation (6) to approximate p(x,y). The authors say that because the latent space is expressive, we can assume (6), but I think this argument is the wrong way round: because of the choice of model (6) and negative log-likelihood loss, the latent z is forced to encode information about y.
- what does ‘top-down’ as in ‘top-down generator model’ mean?
- Which molecular properties are given as regression targets during pretraining, and which are only introduced during SGDS?


**Strength And Weaknesses:**

- As far as I know, this model architecture has not previously been used for this problem, and it gets good results on the presented metrics compared to the presented baselines.
-  Related work is not correctly described. In section 2 '... learn a surrogate function to predict properties, and then use Bayesian optimization' is not an accurate description of the molecular swarm optimization in Winter et al. (2019) nor the simulated design-synthesis-test cycles in Segler et al. (2018).  In 3.2 and 4.2 the authors fail to note that Segler et al. (2018) also generated a very high percentage of valid SMILES strings using an LSTM. It would be a good baseline to include.
- Langevin dynamics is notoriously slow to sample multimodal distributions. Is this a problem for MolEBM?
- What happens to validity, novelty, and uniqueness of generated molecules after SGDS?
- Could the model be evaluated on the GuacaMol benchmarks (Brown et al., 2019)?
- Good scores on the presented metrics may not indicate usefulness for de novo drug design. For example, Figure 2 shows that several of the molecules corresponding to bold numbers in table 4 are not reasonable drug candidates (large cycles etc.).


**Summary Of The Paper:**

The paper presents a model and sampling method for de novo drug design. The model consists of
-  multi-layer perceptron that outputs an 'energy' as a function of a latent vector
- LSTM that generates SELFIES strings conditional on the latent vector
- multi-layer perceptrons to predict molecule properties given latent vectors

The model is first trained on a large set of molecules and their properties. Then, it is fine-tuned to generate molecules with desired properties using a new method that the authors name 'sampling with gradual distributional shifting'.


**Summary Of The Review:**

The paper introduces a new model architecture and sampling strategy for de novo drug design. It gets good results on some metrics, compared to the presented baselines, but I am not sure if the work represents a practically useful advance. The authors do not adequately reference and describe prior work on the same problem.

---

> ### Author Response · Authors · 2022-11-19
> **Reply from Authors (2/2)**
>
> ### Q5: Good scores on the presented metrics may not indicate usefulness for de novo drug design.
>
> **A5** : We agree with you that good scores are not equal to useful molecules. The problem can be partially addressed by multi-objective optimization which we also studied. We wish to point out that designing objectives and optimizing the given objectives are two different (though closely related) problems, and in our work, we focus on the latter. Of course the former is also very important. We have added the following paragraph on this issue in the conclusion and discussion section:
>
> “Another limitation is that, for our optimization method, good property scores  may not translate to useful molecules for drug design. This problem may be partially addressed by multi-objective optimization as studied in this paper. In our work, we focus on optimizing given objectives. Designing good objectives or metrics is an equally or perhaps even more important problem that deserves careful investigation. ”
>
> ### Q6: Clarifications on ‘expressive’ latent space
>
> **A6**: Thanks for the question. We meant expressive latent space prior model. Compared to the unit Gaussian prior in VAE, our latent space prior model is learnable and more flexible. We have removed the sentence about “expressive latent space” from Section 3.3 to avoid confusion.
>
>
> ### Q7: Model assumptions on the relationship among $x$, $y$, and $z$.
>
> **A7**: Thanks for your insightful comment. Following your advice, we have removed the sentence about “expressive latent space”, and write in the revised version that equation (6) is used to approximate the true data distribution of $(x, y)$. A flexible and learnable prior distribution $p_\alpha(z)$ helps make the approximation more accurate compared to a fixed prior distribution of $z$.
>
> Indeed the property value $y$ is a deterministic function of the molecule $x$. However, a machine learning model usually cannot learn the deterministic function exactly, and a random error has to be assumed as approximation, so that $p(y|x)$ is probabilistic. As you pointed out, our equation (6) aims to approximate the data distribution of $(x, y)$, with the help of a learnable prior model of $z$. We have added an explanation of this point in the Appendix (section A.1) of the revised version.
>
>
>
> ### Q8: “What does ‘top-down’ as in ‘top-down generator model’ mean?”
>
> **A8**: In Figure 1 of the paper, the generation direction from $z$ to $x$ is top-down. We have minimized the use of “top-down” in the revised version.
>
> ### Q9: “Which molecular properties are given as regression targets during pretraining, and which are only introduced during SGDS?”
>
> **A9**: The same molecule properties are given as regression targets for both pretraining and SGDS. There are no molecule properties that are only introduced during SGDS but not in pretraining. We have added a comment on this point in Section 3.6 of the revised version.

---

> > ### Author Response · Authors · 2022-12-05
> > **Have we addressed your concerns?**
> >
> > Dear Reviewer,
> >
> > Thank you for reviewing our paper and our reply. We tried our best to address your concerns and added additional experiments.
> >
> > If you have any further questions, please do not hesitate to let us know.
> >
> > Authors.

---

> ### Author Response · Authors · 2022-11-19
> **Reply from Authors (1/2)**
>
> Thank you for your detailed review and valuable suggestions.
>
> ### Q1: Missing related works and baselines: Winter et al. (2019), Segler et al. (2018)
>
> **A1**: Thank you for your valuable inputs about the past literature. We have revised our manuscript and cited these papers. We copied related segments below.
>
> “A number of other efforts have been made to optimize molecular properties with genetic algorithms (Nigamet al., 2020), particle-swarm algorithms (Winter et al., 2019), specialized MCMC methods (Xie et al.,2021).”
>
> “After learning continuous representations for molecules, they are further able to optimize using different methods. Segler et al. (2018) proposes to optimize by simulating design-synthesis-test cycles. Gomez-Bombarelli et al. (2018), Jin et al. (2018), and Eckmann et al. (2022) propose to learn a surrogate function to predict properties, and then use Bayesian optimization to optimize the latent vectors. ”
>
> SMILES LSTM is a strong baseline to generate valid molecules. In Section 4.2, we only include baselines with published results on ZINC250K or MOSES. As you suggested, we evaluated our model on the GuacaMol benchmarks, and compared it to SMILES LSTM in order to have a fair comparison (also see Q4).
>
> ### Q2: Langevin dynamics is notoriously slow to sample multimodal distributions. Is this a problem for MolEBM?
>
> **A2**: This is a good point. Indeed for high dimensional density such as density in image space, Langevin sampling can be slow. However, in our work, $z$ is a relatively low-dimensional vector (100-dimensional), thus the prior and posterior sampling of latent $z$ is highly affordable.
>
> The prior model $p_\alpha(z)$ is a correction (in the form of exponential tilting) of a unit Gaussian density, and in our experience, the Langevin dynamics has no difficulty exploring the distribution. In Figure 5 ( in the Appendix A.5), we show the generated molecules of a long run chain of Langevin dynamics. There is no sign that the Langevin dynamics is trapped in local modes.
>
>
> ### Q3: What happens to validity, novelty, and uniqueness of generated molecules after SGDS?
>
>
> **A3**: Here we use multi-objective ACAA1 binding affinity maximization as an example. After 10 steps of SGDS, the validity, novelty and uniqueness become 100%, 99.94% and 99.31% respectively. After shifting ends, our model is still able to produce molecules with high metrics.
>
> ### Q4: Evaluations on the GuacaMol benchmarks (Brown et al., 2019)
>
> **A4**: Thank you for your suggestion! We are running experiments to evaluate our model with the GuacaMol benchmarks in terms of both unconditional generation and multi-property objective (MPO) optimization (see the added section A.8 in the Appendix). Our preliminary results are displayed in the tables below (also see Table 10 and Table 11 in the Appendix). As you can see, our model, MolEBM, achieves 100% on validity and uniqueness and 99.9% on novelty. For MPO, MolEBM also results in competitive performance to the best model presented in Brown et al., 2019. (Due to the time limit, we did not have time to tune our code for the new experiments. We expect to obtain improved results with fine-tuning.)
>
> | Benchmark  |    AAE    | Graph MCTS | Random Sampler | SMILES LSTM |  VAE  |   MOLEBM  |
> |------------|:---------:|:----------:|:--------------:|:-----------:|:-----:|:---------:|
> | Validity   |   0.822   |  **1.000** |    **1.000**   |  **1.000**  | 0.959 | **1.000** |
> | Uniqueness | **1.000** |  **1.000** |      0.997     |  **1.000**  | 0.999 | **1.000** |
> | Novelty    |   0.998   |    0.994   |      0.000     |    0.912    | 0.971 | **0.999** |
>
> | Benchmark        |  Best of Dataset  | SMILES GA | Graph MCTS | Graph GA | SMILES LSTM  |   MOLEBM  |
> |------------------|:-----:|:----------:|:--------------:|:-----------:|:-----:|:---------:|
> | Osimertinib MPO  | 0.839 |    0.886   |      0.784     |  **0.953**  | 0.907 |  _0.933_  |
> | Fexofenadine MPO | 0.817 |    0.931   |      0.695     |  **0.998**  | 0.959 |  _0.971_  |
> | Ranolazine MPO   | 0.792 |    0.881   |      0.616     |   _0.920_   | 0.855 | **0.924** |
> | Sitagliptin MPO  | 0.509 |    0.689   |      0.458     |  **0.891**  | 0.545 |  _0.829_  |

---

> > ### Comment · Reviewer_vKUH · 2022-12-08
> > **Thank you for the clarifications and additional experiments**
> >
> > Thank you for the clarifications and additional experiments.  In view of these improvements I raised my score to a 5.
> >
> > The equation numbers in some text (‘Estimating expectations in equations 17, 18, and 19’) need to be updated.
> >
> > I understand that the simple SMILES LSTM is not trained on the same datasets as MolEBM, but I still think that the text in 4.2 is misleading: being able to generate a high percentage of valid SMILES strings is not novel.
> >
> > I agree with you that improving property prediction is beyond the scope of the paper, but is it really true that ‘Binding affinity… can be computed with AutoDock-GPU’? Do you have some reference regarding accuracy of the KD estimates from this software? Docking scores are reported to have poor correlation with binding affinity, e.g., https://chemistry-europe.onlinelibrary.wiley.com/doi/10.1002/cmdc.202200425.

---

> > > ### Author Response · Authors · 2022-12-11
> > > **Thank you for your valuable comments**
> > >
> > > Thank you for your further comments and advice.
> > >
> > > (1) We will update the equation numbers. Thanks.
> > >
> > > (2) We will modify the text in 4.2. In particular, we will highlight the results of the prior work of SMILES LSTM.
> > >
> > > (3) We will revise the text about AutoDock-GPU, discussing its limitation as you pointed out. We plan to add the following paragraph:
> > >
> > > We use dissociation constant $K_D$, calculated by AutoDock-GPU as an estimation of binding affinities following steps in (Appendix A.6. in [1]). Binding affinity quantifies the binding strength of a ligand to a protein. Docking score is an extremely rough attempt to approximate binding affinities [3, 4]. The latest AutoDock uses a semiempirical free energy force field to predict binding free energies of small molecules to macromolecular targets [2]. Even though docking is a rough estimate, it is also useful, e.g. [1] used the 5 best-scoring poses from AutoDock-GPU to compute absolute free energy (ABFE), which is calculated by detailed and computationally expensive molecular dynamics (MD) simulations [5]. By comparing experimental binding affinity (requiring enzymes and their existing binders) with in-silico tools, docking methods are case-dependent [6] and system-dependent [7]. For some target proteins, docking scores and binding affinity are not strongly correlated [6,7]. The main reasons we use AutoDock-GPU are as follows. 1) We want our method to be comparable to benchmarks (such as [1]). 2) To study those enzymes without known binders (e.g. ACAA1 studied in our paper), it is not a bad choice to start from docking based methods in terms of computational cost. 3) We aim to provide a general optimization method (sampling with gradual distribution shift, SGDS), which can be applied to any computable property scores.
> > >
> > >
> > > [1] Eckmann, Peter, et al. "LIMO: Latent Inceptionism for Targeted Molecule Generation." ICML (2022).
> > >
> > > [2] Morris, Garrett M., et al. "AutoDock4 and AutoDockTools4: Automated docking with selective receptor flexibility." Journal of computational chemistry 30.16 (2009): 2785-2791.
> > >
> > > [3] Leach, Andrew R., Brian K. Shoichet, and Catherine E. Peishoff. "Prediction of protein− ligand interactions. Docking and scoring: successes and gaps." Journal of medicinal chemistry 49.20 (2006): 5851-5855.
> > >
> > > [4] https://web.stanford.edu/class/archive/cs/cs279/cs279.1222/lectures/lecture11.pdf
> > >
> > > [5] Cournia, Zoe, et al. "Rigorous free energy simulations in virtual screening." Journal of Chemical Information and Modeling 60.9 (2020): 4153-4169.
> > >
> > > [6] Nguyen, Nguyen Thanh, et al. "Autodock vina adopts more accurate binding poses but autodock4 forms better binding affinity." Journal of Chemical Information and Modeling 60.1 (2019): 204-211.
> > >
> > > [7] Breznik, Marko, et al. "Prioritizing Small Sets of Molecules for Synthesis through in‐silico Tools: A Comparison of Common Ranking Methods." ChemMedChem (2022): e202200425.
> > >
> > >
> > > Thanks again for your precious time and valuable advice.

---

### Official Review · Reviewer_gP5J · 2022-10-21

**Confidence:** 3
**Correctness:** 4
**Technical Novelty And Significance:** 2
**Empirical Novelty And Significance:** 2
**Recommendation:** 6

**Clarity, Quality, Novelty And Reproducibility:**

The paper is clear and appears technically solid.
Figure 2 is not referenced in the text.
It is not clear to me what specific top 3 molecules were found in the PlogP experiment.

Source code implementing the method is provided, which is very good! The structure and documentation of the code could be improved.


**Strength And Weaknesses:**

Strenghts

The proposed method is fairly simple and seems to work well.

Weaknesses

The sampling with gradual distribution shifting (SGDS) requires that the properties of interest can be easily computed on the fly. This could be a limiting factor, if the computing the properties cannot be computed quickly. It would be nice with a more detailed discussion of how this (a few comments are included in the appendix).

**Summary Of The Paper:**

The paper presents a method for conditional generation of molecular structures, formulated as a joint latent variable generative model of the molecular structure and one or more properties. The latent variable are modelled using an energy based model, and the molecular structure is an LSTM neural network that generates a SELFIES string representation. Conditionaly sampling molecules with novel values of the desired properties is not likely to work well, since the generative model has not seen any training data in that region. To combat this issue, a procedure is proposed where additional data at the boundary is gradually added, and the model is finetuned.

**Summary Of The Review:**

An interesting paper. Limited technical novelty. Fairly good results.

---

> ### Author Response · Authors · 2022-11-19
> **Reply from Authors**
>
> Thank you for your review and your insightful comments.
>
> ### Q1: The sampling with gradual distribution shifting (SGDS) requires that the properties of interest can be easily computed on the fly. This could be a limiting factor, if the computing the properties cannot be computed quickly. It would be nice with a more detailed discussion of how this (a few comments are included in the appendix).
>
> **A1**: We agree with you, and we have added the following paragraph on this issue in the conclusion and discussion section of the revised manuscript:
>
> “A limiting factor is that the sampling with gradual distribution shifting (SGDS) requires that the properties of interest can be easily computed. In our future work, we shall explore semi-supervised learning methods using our model where the properties are available only for a small sample of molecules. ”
>
> We also would like to point out that the compute requirement of our model is actually lower than that of RL-based methods which are widely used in prior molecule design papers. Both our method and RL-based methods require computing the properties on-the-fly, but our method only requires a small number of iterations (10 to 50 distribution shifting iterations).
>
> ### Q2: The paper is clear and appears technically solid. Figure 2 is not referenced in the text. It is not clear to me what specific top 3 molecules were found in the PlogP experiment.
>
> **A2**: Thanks for the comments. We added a reference to Figure 2 in Section 4.4 of the revised version. We added more plots of the generated molecules, including the p-logP experiment (Appendix A.10), binding affinity experiments (Appendix A.9), in the appendix of the revised version.

---

> > ### Author Response · Authors · 2022-12-05
> > **Have we addressed your concerns?**
> >
> > Dear Reviewer:
> >
> > Thank you for reviewing our paper and our reply.  We have tried our best to address your concerns. If you have any further questions, please let us know.
> >
> > Authors.

---

> > > ### Comment · Reviewer_gP5J · 2022-12-07
> > > **Thank you for the rebuttal**
> > >
> > > Thank you for the rebuttal. I am positive towards the paper, and the main remaining limitation in my view is the limited technical novelty.

---

### Official Review · Reviewer_mgPa · 2022-10-24

**Confidence:** 4
**Correctness:** 4
**Technical Novelty And Significance:** 2
**Empirical Novelty And Significance:** Not applicable
**Recommendation:** 5

**Clarity, Quality, Novelty And Reproducibility:**

Clarity: The paper is well written, but in order to understand some parts of theory one needs to go back to the [Pang2020] paper.

Novelty : The paper is an extension of the  LLSEBprior model [Pang2020] with which parts of the text are shared. This work differs by adding a property regression model on the latent space of the LLSEBprior model which is trained jointly and proposing the gradual distribution shifting (SGDS) to extrapolate the data distribution and sample from the region supported by molecules with high property values. Moreover, the method is applied to molecular data instead of images and text.  The use of the property predictor in the latent space has also used in [Goméz-Bombarelli2018] and [Jin2018] where also is trained jointly as in the proposed model.

Reproducibility: Yes

Pang2020]:  Learning Latent Space Energy-Based Prior Model, NeurIPS 2020
[Gomez-Bombarelli2018]: Automatic chemical de- sign using a data-driven continuous representation of molecules, ACS Cent. Sci. 2018
[Jin2018]: Learning multimodal graph-to-graph translation for molecular optimisation, ICLR 2019


**Strength And Weaknesses:**

Strength:
The paper seems to perform better than state of the art models for unconditional generations using smiles. Especially for unconditional generations it achieves 95.5% validity with 100% novelty using SMILES with the ZINC dataset. Moreover, the sampling with gradual distribution shifting (SGDS) is an interesting suggestion for out of distribution sampling.

Weaknesses:
The paper is clearly written but is missing some more explanation about the sampling procedure and the theory behind it.  Even though to estimate the expirations MCMC sampling of the prior and the posterior is required the method is described as "maximum likelihood”.
Results are not compelling enough. Event though the model is proposed for conditional generations given desired properties in the experimental results we see results only for property optimisation. So more experiments should be added to prove the capability for conditional molecule generation like
- experiments with properties targeted to a predefined range, providing the percent of generated molecules within the target range and the diversity
- experiments aiming to generate new molecules with a optimised property but similar to the original molecules (similarity-constrained Optimisation)

Moreover, a comparison of the generation time of the proposed model with the other models would be useful.

The authors should not bold the best values only when they correspond to your model (Table3 GCPN, MolDQN, MARS, GraphDF achieve same QED 1st with the proposed model, MARS, GraphDF achieve same QED 2nd and 3rd with the proposed model)

**Summary Of The Paper:**

The aim of this paper is the generation of molecules with desired chemical and biological properties learning the joint distribution of molecules and properties . In order to achieve this the authors propose an energy based generative model that augments a top down generative model (conditional autoregressive model) with a latent space learnt via EBM and a property regression model for each of the properties. Then, to sample molecules with desired properties they design an algorithm to shift the learned distribution iteratively.


**Summary Of The Review:**

The paper extends the LLSEBprior model with a property predictor on the latent space in order to generate molecules with desired chemical and biological properties.
The contribution of the paper is limited and more experiments are needed in order to illustrate the capability of the proposed model to generate molecules with specific properties (see Weaknesses).

---

> ### Author Response · Authors · 2022-11-19
> **Reply from Authors**
>
> Thank you for your review and your valuable comments.
>
> ### Q1: The paper is clearly written but is missing some more explanation about the sampling procedure and the theory behind it. Even though to estimate the expectation MCMC sampling of the prior and the posterior is required the method is described as "maximum likelihood”.
>
> **A1**: Thanks for your comment. We have added a section in Appendix (section A.1) to provide details about the derivations of the equations in Section 3.4. In the Appendix, we also explain that the model is learned by maximum likelihood where the gradient of the log-likelihood involves expectations with respect to the prior and posterior distributions of the latent vector. The expectations are approximated via Monte Carlo sampling, so that the learning algorithm is stochastic gradient ascent, or stochastic approximation algorithm whose convergence was established by Robbins and Monro [1]. In the case of Langevin sampling, the bias caused by the finite step MCMC was analyzed by Pang et al. (2020). In the revised version, we refer to the learning method more carefully as approximate maximum likelihood.
>
> [1] Robbins, H., & Monro, S. (1951). A stochastic approximation method. The annals of mathematical statistics, 400-407.
>
> ### Q2: Additional experiments on similarity-constrained optimisation
>
> **A2**: Thank you for your advice. We have conducted experiments following your suggestion and the preliminary results are shown in Appendix A.7. In the logP targeting experiments (see A.7.1 and Table 8), MolEBM is able to get strong diversity scores with significantly better success rate. In the similarity-constrained p-logp optimization experiments (see A.7.2 and Table 9), MolEBM achieves the highest improvement percentage and success rate at $\delta=0$ and $\delta=0.2$ and competitive results when $\delta=0.4$. Due to the time limit, we did not have time to tune our code for the new experiments. We expect to obtain improved results with fine-tuning.
>
> | Method     | -2.5<=logP<= -2 | -2.5<=logP<= -2 | 5<=logP<= 5.5 | 5<=logP<= 5.5 |
> |------------|-----------------|-----------------|---------------|---------------|
> |            | Success         | Diversity       | Success       | Diversity     |
> | ZINC       | 0.4%            | 0.919           | 1.3%          | 0.901         |
> | JT-VAE     | 11.3%           | 0.846           | 7.6%          | 0.907         |
> | ORGAN      | 0               | -               | 0.2%          | **0.909**     |
> | GCPN       | 85.5%           | 0.392           | 54.7%         | 0.855         |
> | LIMO       | 10.4%           | **0.914**       | -             | -             |
> | **MolEBM** | **86.0%**       | 0.874           | **62.2%**     | 0.858         |
>
> | $\delta$ |     GCPN    |   GCPN  |     GraphDF     |  GraphDF  |     LIMO     |   LIMO  |      MolEBM    |   MolEBM  |
> |----------|-----------|-------|---------------|---------|------------|-------|------------------|---------|
> |        |   Improv.   |  %Succ. |     Improv.     | %Succ. |    Improv.   |  %Succ. |       Improv.      | %Succ. |
> | 0        | 4.2$\pm$1.3 | **100** |   5.9$\pm$2.0   |  **100**  | 10.1$\pm$2.3 | **100** | **19.11$\pm$2.12** |  **100**  |
> | 0.2      | 4.1$\pm$1.2 | **100** |   5.6$\pm$1.7   |  **100**  |  5.8$\pm$2.6 |   99.0  |  **7.41$\pm$1.89** |  **100**  |
> | 0.4      | 2.5$\pm$1.3 | **100** | **4.1$\pm$1.4** |  **100**  |  3.6$\pm$2.3 |   93.7  |    3.80$\pm$1.44   |    97.5   |
>
>
>
>
> ### Q3: Generation time comparison.
>
> **A3**: Thanks for the advice. We have added the table below for comparison (also see Appendix A.6).
>
> | Model   | Penalized-logP/QED |
> |---------|:------------------:|
> | JT-VAE  |         24h        |
> | GCPN    |         8h         |
> | MolDQN  |         24h        |
> | GraphDF |         8h         |
> | Mars    |         12h        |
> | LIMO    |         1h         |
> | MolEBM  |        4.5h        |
>
>
> ### Q4: Bold font on the best models.
>
> **A4**: Thanks for your advice. In the revised version, we bold all the best values in Table 3 following your advice.
>
> ### Q5: Clarity on the technical details.
>
> **A5**: Thanks for your feedback. To make the paper self-contained, we have added a section in the appendix (section A.1) about the technical details of the learning method.
>
> ### Q6: Clarification on the novelty.
>
> **A6**: Thanks for your comments. Our work is the first to apply the latent space EBM prior model to molecule design. Our optimization method based on sampling by gradual distribution shifting is novel, and is potentially useful for property optimization. We have cited [Goméz-Bombarelli2018] and [Jin2018] in the revised version.

---

> > ### Author Response · Authors · 2022-12-05
> > **Have we addressed your concerns?**
> >
> > Dear Reviewer:
> >
> > We tried our best to address your concerns. If you have any further questions, please let us know. Thank you for reviewing our paper and our reply.
> >
> > Authors.

---

### Decision · Program_Chairs · 2023-01-20

**Decision:**

Reject

**Justification For Why Not Higher Score:**

The weaknesses of the paper that I have listed above are crucial and should be addressed for publication of this paper. I recommend addressing these issues, that is, more careful examination of the novel technique in the proposal compared to (Pang et al., 2020), improving presentation of the paper, and thorough experiments on the GuacaMol benchmark.


**Justification For Why Not Lower Score:**

N/A

**Metareview: Summary, Strengths And Weaknesses:**

This paper proposes a molecular generation approach using the energy based model (EBM). The task of molecular generation is recently actively studied in machine learning and chemoinformatics, and this paper addresses this problem by the well-motivated EBM-based approach. The proposed method learns a joint distribution of molecules and the target property using the EMB, and generates new molecules by MCMC sampling from the latent space represented by the trained model.

### Strength

- The proposed approach is simple and flexible. In particular, it can directly incorporate multi-objective optimization, which is an important task in molecular generation.
- Presentation is overall good. The paper is clearly written and easy to follow.

### Weakness

- As reviewers pointed out, the key concept and technique of this work is common to (Pang et al., 2020), with an extension of the regression part and its application to molecular generation. Therefore the novelty of the technical contribution is not high. I understand that the authors newly propose the SGDS algorithm to effectively treat the regression part in sampling. However, the contribution of SGDS itself compared to (Pang et al., 2020) is not carefully examined. The ablation study is not so informative as it is known to be relatively easy to optimize the Penalized logP (more carbon and less hydroxyl groups).

- Although the presentation of this paper is good, one can easily find many text overlaps with (Pang et al., 2020). I recommend rewriting such parts using the authors' own words.

- Experiments are not thorough. Although the authors have added experiments on the GuacaMol benchmark, which I acknowledge, unfortunately it is not complete. Evaluation on the KL divergence and the FCD is missing, and more molecules should be taken into account in the target molecular re-generation experiment. Since GuacaMol is currently used as the standard benchmark, such experiments are required in empirical evaluation of the proposed approach.